# Preserving the spots: Jaguar (*Panthera onca*) distribution and priority conservation areas in Colombia

María Camila Machado-Aguilera[1,2], Leonardo Lemus-Mejía[1], Jairo Pérez-Torres[2], Diego A. Zárrate-Charry[1,3], Andrés Arias-Alzate[4], José F. González-Maya[1,5]*

1 Proyecto de Conservación de Aguas y Tierras – ProCAT Colombia, Bogotá, Colombia, 2 Laboratorio de Ecología Funcional, Unidad de Ecología y Sistemática (UNESIS), Depto. Biología, Pontificia Universidad Javeriana, Bogotá, Colombia, 3 WWF Colombia, Bogotá, Colombia, 4 Facultad de Ciencias y Biotecnología, Universidad CES, Medellín, Colombia, 5 Departamento de Ciencias Ambientales, División de Ciencias Biológicas y de la Salud, Universidad Autónoma Metropolitana Unidad Lerma, Lerma de Villada, Estado de México, México

* jfgonzalezmaya@gmail.com

**Data Availability Statement:** All relevant data are within the manuscript and its Supporting information files and through Open Science

## Abstract

The jaguar (*Panthera onca*) is a charismatic species considered Vulnerable in Colombia but yet largely unknown in the country. The species is mostly threatened by the continuous decline in its habitats, mostly derived from deforestation and habitat loss, additional to hunting and conflicts with humans. Thus, the future of jaguars in Colombia depends on protecting and recovering existing habitats. The aims of this study were to 1) evaluate jaguar distribution and identify the remnant patches of habitat in Colombia, 2) define an ecological connectivity network within the country, and 3) propose a priority areas portfolio for the conservation and recovery of jaguars. We used a presence background model for estimating species potential distribution and subsequently identified remaining habitat patches across the country based on land cover and species-specific ecological attributes. We then created an ecological connectivity network based on circuit theory and following a multi-criteria approach identified jaguar priority areas for conservation (JPCA) and recovery (JPRA). Jaguar potential distribution comprises 1103122.43 km², from which 56.71% maintain suitable patches of potential habitat. We identified 960 corridors between remnant patches of natural or semi-natural vegetation. Based on the criteria, JPCAs with greater importance were identified in each of the five Colombian regions. JPRAs were located mainly towards the Andean and Caribbean regions. These JPCAs and JPRAs could serve as a guide for designing and implementing management strategies for the long-term conservation and recovery of the species in Colombia.

## Introduction

Land-use change is the main driver for habitat loss and fragmentation, the leading causes of biodiversity loss around the world [1, 2]. In Colombia, these processes result from constant land transformations generated by an increase in urbanization, cattle ranching, agriculture,

Framework online repository (1. Gonzalez-Maya JF, Machado-Aguilera MC. Jaguar (Panthera onca) distribution records for Colombia. OSF; 2024. doi:10.17605/OSF.IO/RKDS9).

**Funding:** The author(s) received no specific funding for this work.

**Competing interests:** The authors received no specific funding for this work.

and illicit crops [3]. This has caused changes in landscape structures, with detrimental effects on species ecological dynamics (especially for carnivore species), such as population isolation, genetic flow alteration, changes in abundance and distribution, among others [1, 4, 5].

The jaguar (*Panthera onca*) is a predator classified as Near Threatened (NT) globally according to the IUCN Red List of Threatened Species [6] but considered Vulnerable (VU) in Colombia [7] and in most countries of its distribution, and in spite of such discrepancy, and most importantly, it is estimated that ~97% of its populations are considered either Endangered or Critically Endangered at continental scale [8]. It is the largest felid in the Americas and the only representative of the genus *Panthera* in the continent [9, 10]. Its overall requirements of extensive areas of natural habitats and high prey availability [11, 12] and low reproductive rates and densities [13–15], make it a susceptible species and an indicator of healthy ecosystems [13]. The species is found in a great variety of habitats, from dense forests to swamps or dry areas [16], and is mainly found in elevations below 3200 m [17, 18].

Colombia has been recognized as a critical region for connecting jaguar populations between Central and South America [19]. Within the country, five jaguar subpopulations have been identified; the Amazon, Biogeographic Choco, Paramillo-San Lucas, Sierra Nevada de Santa Marta and Serrania de Perija-Catatumbo [8]. However, the historical landscape changes caused by habitat fragmentation and loss, are decreasing the species potential habitat [20–22] and have reduced its occupied distribution by 39% in the country [23]. Consequently, the species populations are increasingly isolated, and its genetic diversity is reducing, making the jaguar more vulnerable to extinction at different scales [24]. Additionally, local communities' excessive hunting of their prey increases the negative interactions with humans due to cattle predation and retaliatory hunting, which enhances the risk for the species [21, 22, 25, 26].

In response to these threats, previous efforts for identifying priority areas and corridors for the species conservation have occurred at continental [14, 18, 19] and national levels, like Mexico [10, 27], Brazil [28, 29], and Nicaragua [30]. In Colombia, previous approximations have identified some priority areas [23, 31, 32], and only one has proposed potential jaguar corridors at national level [31]. Nevertheless, considering the availability of new tools for generating objective spatial approaches, the considerable increase in basic jaguar information, and the need for robust and comprehensive landscape analysis for the species in a critical area such as Colombia, generating new approaches for spatially assessing jaguar conservation priorities was warranted. The objectives of this study where to: 1) evaluate jaguar distribution and identify the potential remnant habitat patches in Colombia, 2) define the ecological connectivity network within the remnant patches, and 3) propose a portfolio with prioritization areas of potential habitat remnants for the conservation and recovery of the jaguar in Colombia.

## Materials and methods

### Study area

Colombia is considered the critical connection between jaguar populations of Central and South America [18]. It is located in the northwestern part of South America, sharing borders with Panama, Venezuela, Ecuador, Peru, Brazil, the Caribbean Sea, and the Pacific Ocean [33]. It has a land surface extension of 1141748 km$^2$ [34] and comprises five natural regions: Amazon, Andean, Caribbean, Orinoco and Pacific [34, 35].

The country's topography is dominated by the presence of the Andean Mountain range (i.e., the Andes) which splits into three independent branches (i.e., central, eastern, and western cordilleras) and an isolated range, the Sierra Nevada de Santa Marta. This topography also includes the presence of unique biomes such as the Orinoco plains, locally known as "Llanos Orientales" and the Amazon rainforests [36]. These large orographic and altitudinal variations

(0–5800 m) determine the high environmental variability of the country [36, 37], which generates a high ecosystems diversity, allowing the presence of a great number of species, making Colombia one of the most biodiverse countries of the world [37, 38].

Colombia has a large variety of ecosystems, from deserts to tropical forests [37], having a total of 91 ecosystems, of which 70 are natural and 21 transformed [39]. Nevertheless, 22 ecosystems are categorized as critically endangered, 14 as endangered, and 12 as vulnerable [37]. Cattle ranching is the country's largest economic activity with the highest representation, occupying 85% of the agricultural lands, reason why it is considered the dominant land-use of the country and the main driver of landscape transformation [3].

The country also has 498745.15 km$^2$ covered by protected areas, including terrestrial and marine areas. In total, as for 2020, Colombia had 1671 protected areas from which 1236 are local protected areas, 313 regional and, 122 national-level areas [40]. New areas are either under planning or in the process of official declaration, especially at sub-national levels and in private lands, and most likely this number will keep changing in coming years.

## Species data

As the basis for the modeling approach, we collected as many available jaguar records as possible for the country from multiple sources. We collected records of jaguar presence from secondary sources such as 1) biodiversity data sets (i.e., GBIF, Species Link, Data Basin [41–43], 2) literature [44–54], and 3) our own data derived from validated records of direct observations, camera-trap data, predation events with confirmation of the predator, among others. We applied two complementary filters for depurating the records to obtain the most reliable species potential distribution. The first filter focuses on credibility and the second focuses on geographic precision of each record [55]. Credibility combines the type of evidence (e.g., human observation, machine observation, collected specimen, etc.) with source type (e.g., scientific articles, thesis, etc.). Geographic precision is based on the spatial information associated with each record and validated with official cartographic sources [55]. We carried out this process using an RStudio code [56–64]. According to these filters (i.e., credibility and geographic precision), we classified each record in three categories of reliability: low, medium, or high (Table 1) [65]. We removed all records classified as with low reliability and dropped duplicated

**Table 1. Criteria used for classifying reliability of *Panthera onca* records for distribution modelling of the species in Colombia.**

| Criteria | Attributes | Reliability |
|---|---|---|
| **Evidence** | Preserved specimen | High |
| | Machine observation | High |
| | Human observation | Medium |
| | Material sample | Medium |
| | No data | Low |
| **Source** | Peer reviewed article | High |
| | Expert validates record | High |
| | Museum | Medium |
| | Datasets | Medium/high |
| **Geographic precision** | Department and Municipality | High |
| | Department | Medium |
| | Municipality | Low |
| | None | Low |

information from the analysis. Finally, we applied a spatial thinning of 1km for all other records [66].

## Calibration area and climatic data

To generate the species potential distribution, we first identified and selected all ecoregions [67] with historical and current confirmed records of the species and defined these as the accessible area (M *sensu stricto*) [68] and the model´s calibration area [69, 70]. Afterwards, we obtained 19 bioclimatic variables from WorldClim2 [71] and estimated Spearman correlation scores in order to remove highly correlated variables, with values equal to or greater than 0.8 [70]. We also considered the most important variables according to permutation and contribution percentages, based on the jackknife test implemented in Maxent [72] and the variables used in other studies [28, 70]. We then selected seven bioclimatic variables: mean diurnal range (Bio2), temperature seasonality (Bio4), max temperature of warmest Month (Bio5), temperature annual range (Bio7), annual precipitation (Bio12), precipitation of driest month (Bio14), and precipitation of coldest quarter (Bio19).

## Jaguar potential distribution estimation

With the selected variables, we modeled the species potential distribution using Wallace platform [73], based on the maximum entropy (Maxent) algorithm [69]. For parametrization and calibration, and to obtain the best possible model that better fitted the data, we used five feature classes (linear = L, hinge = H, linear/quadratic = LQ, linear/quadratic/hinge = LQH and linear/quadratic/hinge/product = LQHP) [74–76] and nine regularization multipliers (0.5, 1, 1.5, 2, 2.5, 3, 3.5, 4 and 4.5) [72, 75]. Then from all possible candidate models, we selected the best model using the Akaike information criterion (AIC); the best model was the one with the lowest AIC value [77, 78]. This model was validated using a spatial partitioning approach using the checkerboard2 method [77]. Subsequently, the final model was transformed into a binary output using the minimum training presence threshold as it reduces omission errors [79]. Then we refined the model based on the altitude range in which the jaguar has been reported (0–3200 m) [16, 17], and estimated the total range currently covered by protected areas in the country.

## Identification of remaining habitat patches

We used a spatial approach using the national land-cover types via GIS procedures to identify the remnant patches of potential habitat for the species within its estimated distribution range. We obtained the national land-cover map for 2012 [80] and selected the land-covers reported as jaguar habitat from the literature: dense forest, open forest, gallery forest, fragmented forest, secondary vegetation, natural grasslands, swamps, and coastal swamps [41, 70, 81, 82]. We then clipped these coverages with the species potential distribution to obtain a proxy of the potentially occupied habitats. Then we estimated each patch area and selected those with a size equal to or greater than 23.5 km$^2$, based on previous home range estimations, but that might serve as stepping stones that would allow movement through fragmented landscapes [29].

## Functional connectivity network definition

We created the connectivity network based on circuit theory [83]. For this, we used remnant patches identified and developed a resistance layer based on the human footprint index for Colombia (HFI) [84]. This layer is composed of variables that better reflect human influence (i.e., rural population density, distance to roads and settlements, land use, fragmentation index

of natural vegetation, biomass index, and time of intervention) [84]and the ones that most likely restrict the species mobility across a landscape. We rescaled the human footprint index from 1–100 to create a resistance layer [85], where the highest values represented a greater human influence [84] and hence a greater resistance for the species mobility. For this analysis, we used Linkage Mapper 2.0 [86] toolbox in ArcGIS to estimate the least cost path corridors between patches. A maximum Euclidean distance of 113 km was chosen, which is the median dispersal distance of the species [29], thus preventing corridors overestimation [85]. Finally, we calculated centrality values for each core habitat indicating their importance and contribution to the connectivity network [87–89].

### Selection of jaguar priority areas in Colombia

To define priority areas, we selected the most appropriate criteria for selecting important jaguar areas based on a literature review in online repositories (i.e., Google Scholar, Web of Science). We identified a set of criteria used by multiple similar exercises [14, 28, 29, 90–92] and narrowed them to the most appropriate list according to data availability and pertinence for Colombia. Therefore, we estimated a priority scheme considering four criteria for each core habitat patch: 1) human footprint index (HFI) [84], calculated as the percentage of pixels with HFI values under 15 [84], as a proxy of the overall human influence indicating low human impact [84]; 2) patch size [27–29], as the total area of core habitat; 3) level of protection, as the percentage of each patch under any Protected Area category [91–93]; and, 4) connectivity importance (i.e., centrality value), estimated as the relative importance of each core patch for the whole connectivity network [87–89].

We normalized all criteria values to a 0–1 scale and weighted each of the remnant patches identified. Considering the area requirements of the species [10, 11], we additionally assigned more weight to the area criterion by assigning three scores for patch size: a score of 1 for those patches with areas between 23.5 and 1200 km$^2$ (i.e., one of the largest home ranges reported for a male) [94], score of 2 for areas between 1200 and 5000 km$^2$ and score of 3 for areas greater than 5000 km$^2$ (i.e., areas large enough for maintaining viable a population of 50 individuals) [29].

Then, we defined those areas with total weighted values greater than 3, considering all criteria, as jaguar priority conservation areas (JPCA). We classified these JPCAs into two categories: 1) (JPCA I) areas with values greater than 4 and 2) (JPCA II) areas with weighed values greater than 3 but less or equal to 4. At the same time, we also defined areas with total values less or equal to 3 as jaguar priority recovery areas (JPRA). Namely, the areas with lower weights but greater conservation attention for the species recovery. We classified these JPRAs into two categories: 1) (JPRA I) areas with values greater than 2 but less or equal to 3, and 2) (JPRA II) areas with total weighted values less or equal to 2.

## Results

### Jaguar potential distribution estimation

We obtained 535 confirmed records of the species (S1 Table, also available through the Open Science Framework repository; doi: 10.17605/OSF.IO/RKDS9), of which 388 remained after the filtering process. The best two models (AIC of 10753.6 and 10754.05) had the LQHP feature class, with 4 and 3 regularization multipliers and an omission rate of 0 and 0.005, respectively (S2 Table). According to the permutation importance, the most important predictor variables were precipitation of the driest month (25.6%) and precipitation of the coldest quarter (22.8%), followed by annual precipitation (14.9%) and the temperature annual range (11.4%) (Table 2).

**Table 2. Climatic variables used to estimate the jaguar (*Panthera onca*) potential distribution in Colombia and their permutation importance.**

| Variable | Permutation importance |
|---|---|
| Bio14: Precipitation of Driest Month | 25.65 |
| Bio19: Precipitation of Coldest Quarter | 22.76 |
| Bio12: Annual Precipitation | 14.89 |
| Bio7: Temperature Annual Range | 11.39 |
| Bio2: Mean Diurnal Range | 10.58 |
| Bio5: Maximum Temperature of Warmest Month | 9.20 |
| Bio4: Temperature Seasonality | 5.53 |

Total estimated jaguar potential distribution covers an area of 1103122.43 km$^2$ in Colombia, being potentially distributed in almost the entire country, with a very low probability of occupying areas above 3200 m in the three mountain ranges, the Sierra Nevada de Santa Marta, and in a central portion of the Chocó region (Fig 1). Of this range, only 172975.35 km$^2$ (15.6%) are currently under a protection category, consisting mainly of national natural parks (66.18%), natural reserves (11.47%), regional integrated management districts (11.24%), among others.

## Identification of remaining habitat patches

We identified 497 remnant patches of potential habitat, covering a total area of 625532.48 km$^2$, representing 56.71% of the species potential distribution. Of the total area of remaining patches, 367005.31 km$^2$ (58.67%) corresponded to the Amazon, 136518.22 km$^2$ (21.82%) to the Orinoco, 58867.49 km$^2$ (9.41%) to the Andean, 50823.81 km$^2$ (8.12%) to the Pacific and, 12317.65 km$^2$ (1.97%) to the Caribbean regions. Patch sizes ranged from 23.5 to 196466 km$^2$, with a mean (±SD) area of 1258.6±10264.45 km$^2$. The small patches (23.5–100 km$^2$) were the most abundant (72%) and only 7.24% of the patches had sizes larger than 1200 km$^2$ (Fig 2A). The largest patch (196466 km$^2$) is located in the Amazon region and represents 31.4% of the total patches area. The dominant land cover within all patches identified corresponds to dense forest (78.2%), mostly located in the Amazon and Pacific regions, followed by natural and artificial grasslands (18.4%) with a large representation in the Orinoco, and open forest and coastal swamps with the lowest representation with 0.18% and 0.02%, respectively (Fig 2B).

## Functional connectivity network definition

We identified 960 potential corridors between the remnant habitat patches (Fig 3). Of the 497 patches, 439 (88.3%) had at least one corridor, resulting in 58 core habitats isolated. Corridors mean length distance (±SD) was 12.43±19.39 km, ranging from 0.31 to 124.93 km (Fig 3) between core habitat patches. Andean and Caribbean regions showed the largest number of corridors identified and the longest distances (Fig 3). Mean (±SD) number of corridors between patches was 2.19±1.68, with a maximum of 24 corridors for a single remnant patch. Centrality values for all remnant patches varied considerably, with a mean (±SD) value of 5962.31±7378.13, and a maximum value of 45043.7 (Fig 3).

## Selection of jaguar priority areas in Colombia

Our methodological approximation allowed us to identify jaguar priority conservation areas (JPCA) and jaguar priority recovery areas (JPRA; Fig 4) based on four criteria: mean (±SD)

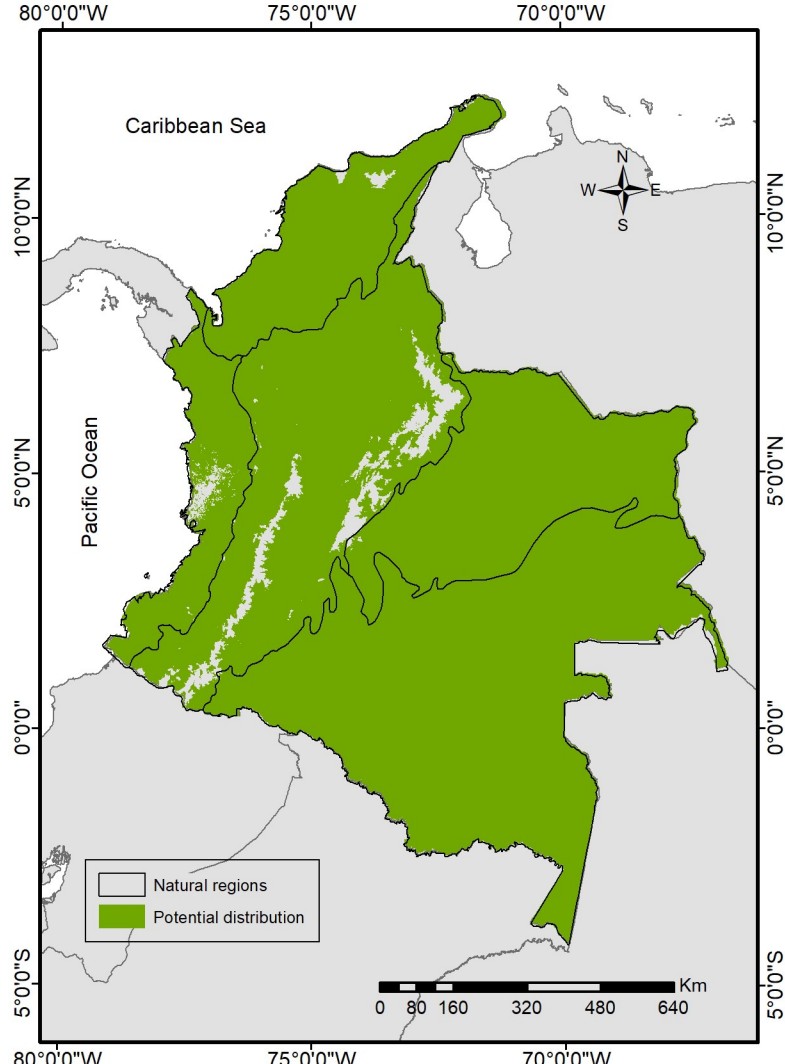

**Fig 1. Jaguar (*Panthera onca*) potential distribution based on an ecological niche modeling approach for Colombia.** Boundaries were obtained from the Vector Map Level 0 [95], and the natural regions from IGAC [35].

patch size was 1258.6±10264.45 km$^2$, where small patches were more frequent; centrality values showed a mean (±SD) index of 5962.31±7378.13; mean (±SD) number of "natural" (<15) HFI pixels was 11130±99946.08, while mean (±SD) proportion of natural pixels was 0.32±0.35; and, total area protected was 137559.31 km$^2$, representing 22% of the total area of remnant patches (Fig 5). The Amazon region had the most significant proportion of the total protected area (70.11%), followed by the Andean (13.11%), Orinoco (9.60%), Caribbean (3.65%) and, Pacific (3.53%) regions (Fig 5). Of the total 497 patches, only 188 (37.8%) have any level of protection, 34 patches (predominantly of small size) had 90–100% of their area protected, and the largest remnant patches (196465.8 and 74583.3 km$^2$) have only 35.1 and 20.4% of its area under protection respectively.

Our prioritization portfolio comprised 34 JPCAs and 463 JPRAs occupying a total area of 560047.06 km$^2$ and 65485.41 km$^2$ respectively. Of these JPCAs, 15 were JPCA II and 19 JPCA

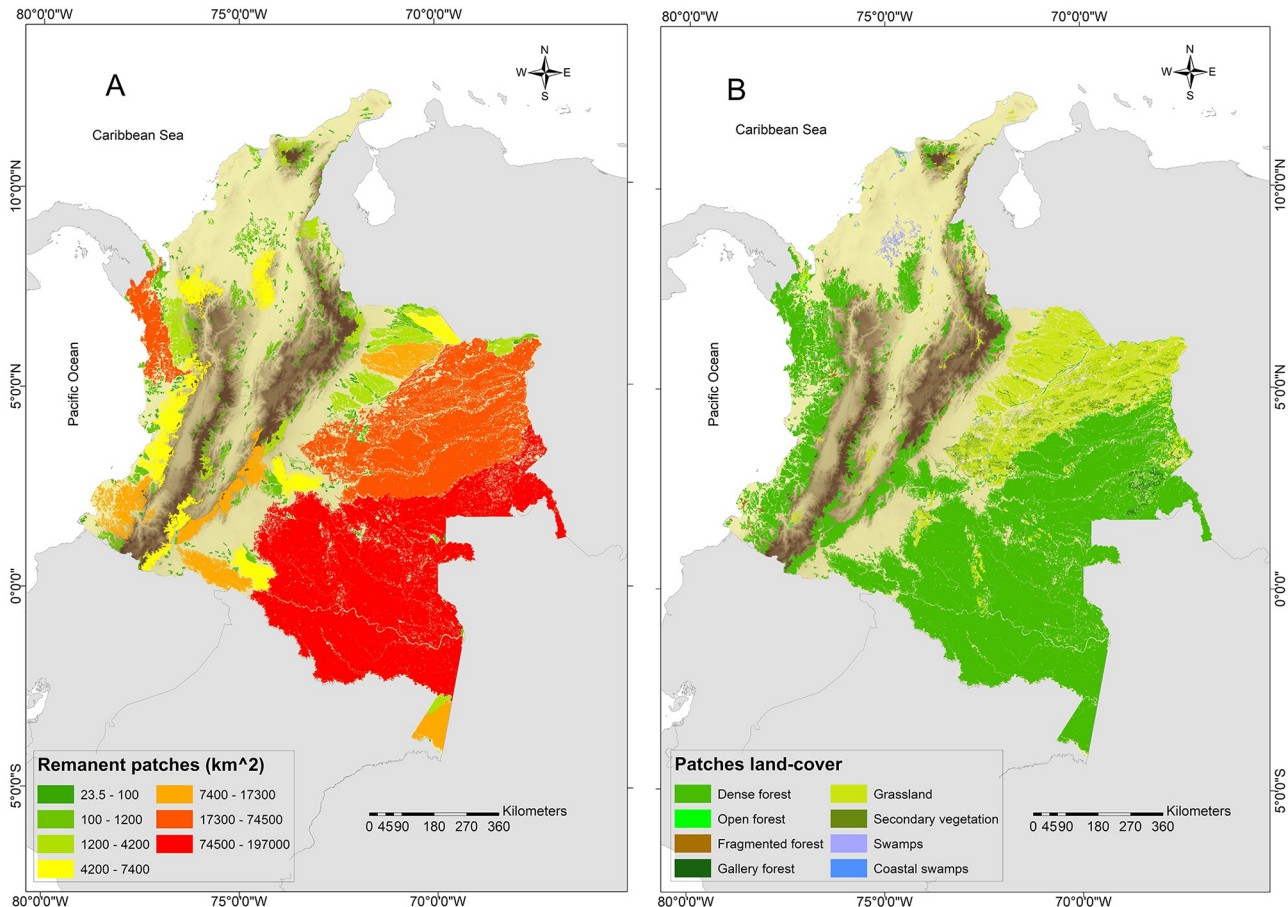

**Fig 2. Distribution of remnant habitat patches for jaguars in Colombia.** (A) Remnant patches classified by size and (B) remnant patches classified by land-cover type. Elevation data obtained from WorldClim2 [71].

I, predominantly found in the Amazon and Orinoco regions (Fig 4). On the other hand, from the JPRAs identified, 377 were JPRA II and 86 JPRA I, with JPRA II mainly located in the Caribbean region and the Andes piedmont, while JPRA I meanly located in the Orinoco region and the inter-Andean valleys of Central Colombia (Fig 4).

## Discussion

Our results indicated that jaguar potential distribution covered a large portion of the Colombian territory (96.62%), however, and due to habitat loss and fragmentation, currently only 56.71% of its distribution corresponds to potential habitat for the species, indicating a reduction of 43.29%, a value 4.29% larger than the reported in other studies [23]. Although it is not possible to know an exact time-frame in which this reduction occurred, previous works suggest that, considering regional heterogeneity, at least 60% of natural covers have disappeared in the last 400 years [3], but it is estimated that since the 1970´s, human influence, namely human footprint, has increased in at least 50% [84]. Though, compared to the potential distribution reported in previous studies [32] our distribution was 227 284.43 km$^2$ greater, which may be due to the fact that in this study the elevation range was considered from 0–3200 m, while previous studies considered 0–2000 m [23, 32].

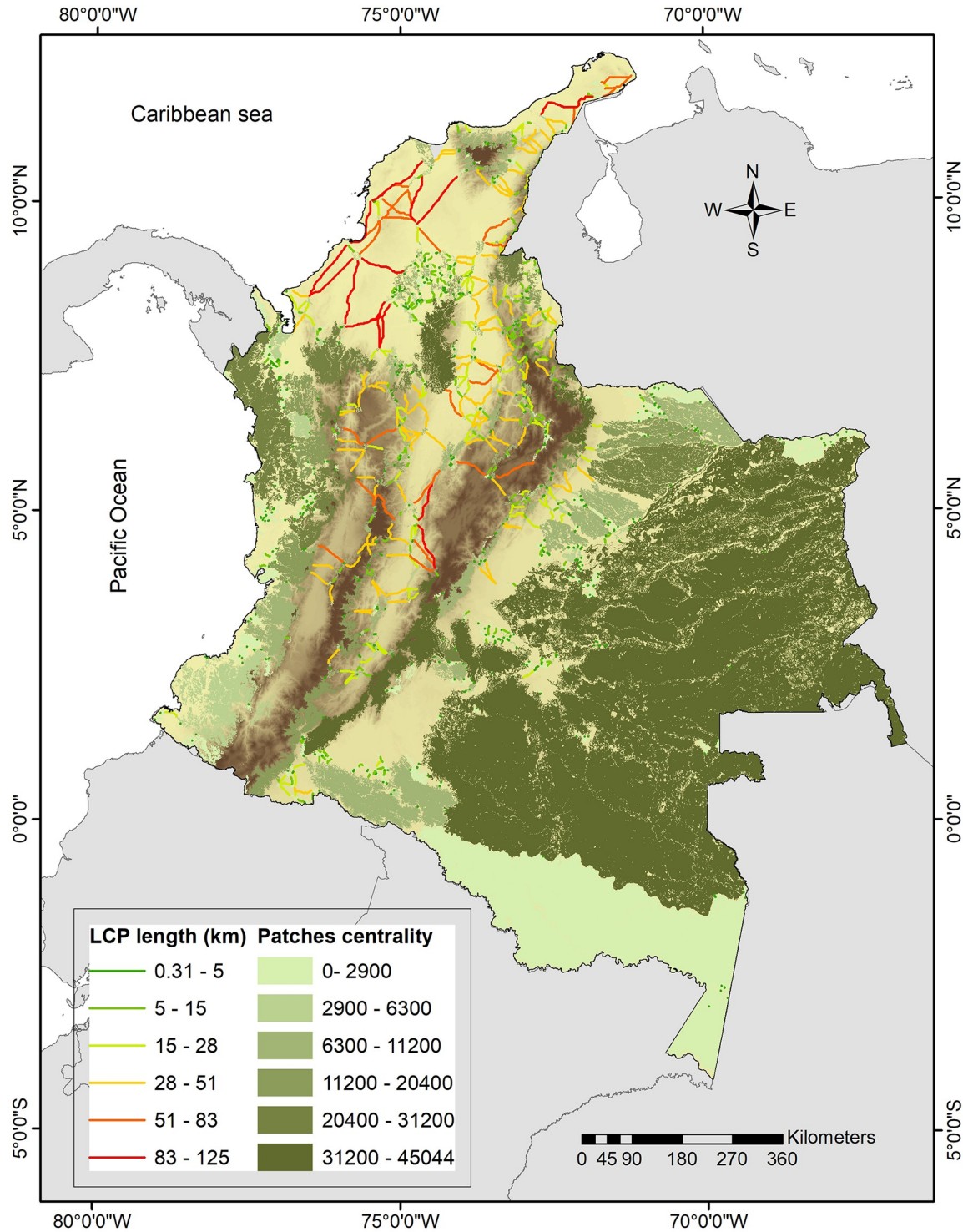

**Fig 3. Potential jaguar corridors distribution in Colombia.** Corridors depicted as least cost path corridors (LCP) classified by length and centrality value (the importance of each patch within the connectivity network) of each remnant patch. Elevation data obtained from WorldClim2 [71].

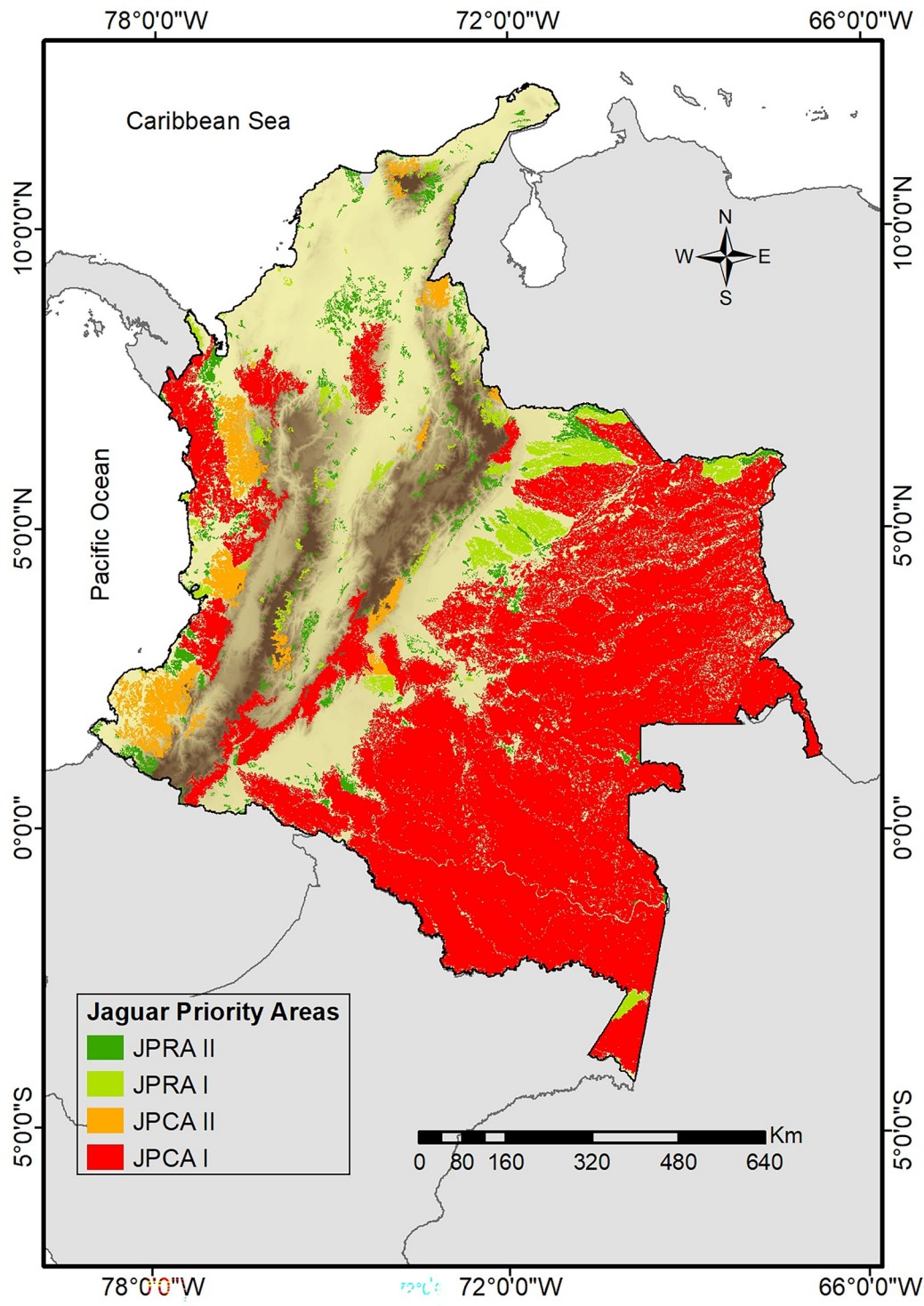

**Fig 4. Distribution of jaguar priority conservation areas for conservation and jaguar priority recovery areas in Colombia.** Priority areas for the conservation type I (JPCA I) and type II (JPCA II), and priority recovery areas type I (JPRA I) and type II (JPRA II). Elevation data obtained from WorldClim2 [71].

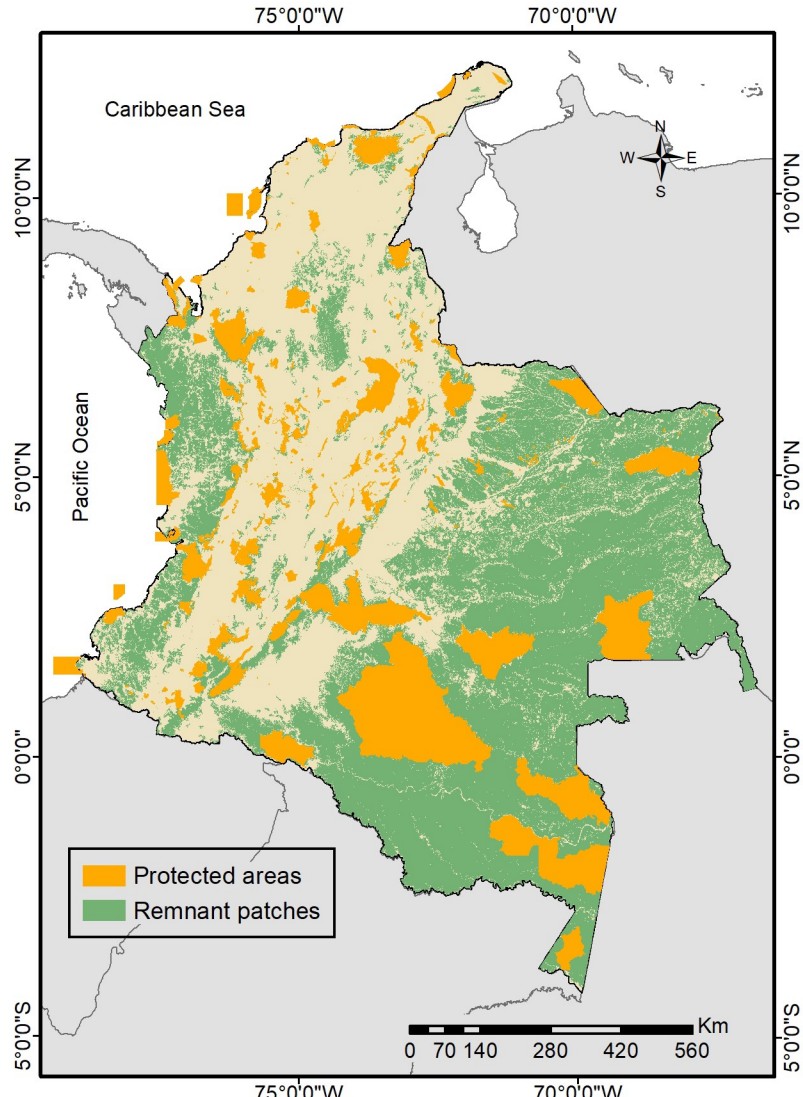

**Fig 5. Protected areas of Colombia within the remnant patches of the jaguar.**

Although the Amazon and Orinoco regions hold the largest remaining habitat extensions for the species, both also face drastic landscape transformations due to persistent human encroachment that have increased in recent years [84]. According to our results, these regions harbor the largest remaining patches with lower human influence and with the largest contribution to national connectivity, thus representing the most important JPCAs in the country. Even though these areas are largely consistent with the Jaguar Conservation Units (JCU) proposed in previous studies [31], our JPCAs incorporate transformation processes, indicating not only the importance of these areas but also their status and are supported on a criteria-based systematic approach that allows monitoring and designing appropriate conservation actions. It is important to highlight that the Orinoco region also hold areas with JPRAs I towards the Andes piedmont, which correspond to areas with historically high human footprint [84].

Comparatively, the Andean region also harbors important and considerably large remaining patches, mostly towards the southern portion of the region, while smaller and isolated patches dominate the northern and central portions of the most populated region in the country. These patches are immersed in highly transformed landscapes, generated by historical population growth, urban settlement concentration, inadequate land use, and large-scale historical expansion of illicit crops [96, 97]. Additionally, the region has JPCAs I and II in the Serranía de San Lucas, the Catatumbo and, the upper Magdalena, which represent areas with historical jaguar presence [8, 31]. For instance, the Catatumbo JPCA II has been identified as an important area for jaguar populations [8], considering its central importance for the connectivity between the Orinoco and Caribbean regions [31]. Meanwhile, Serranía de San Lucas JPCA I has been previously identified as an area of great importance for connecting populations between north and south of the country [32, 98].

The Pacific region, although still retains an important proportion of ecosystems [97], it was identified as the second region with less potential habitat for the species. This might be explained by intensive deforestation processes within the region, mostly associated with illegal mining and wood exploitation [97, 99]. Nevertheless, this region is still an important area for jaguar conservation as shown herein, with important JPCAs and JPRAs all over the territory, consistent with previous conservation approaches [8, 23, 31, 41]. However, it seems the number of jaguars hunted have recently increased exponentially in the region [54] and human-jaguar conflicts have also growth considerably [25]; previous efforts have emphasized the need for conflict assessments and mitigation in the area [21, 100]. Notwithstanding, still these conservation areas are of critical importance for the country, but especially for the continent since they represent the connection between South and Central American populations [8, 25].

Finally, the Caribbean region has the smallest extent of potential habitat in the country, as a result of historical and intensive human production activities (i.e., agricultural and cattle expansion) and the proliferation of human settlements across the region [84, 101]. Nevertheless, still, important habitat patches remain in Sierra Nevada de Santa Marta (SNSM) and Paramillo National Natural Parks, although most of them isolated, thus reducing potential jaguar dispersion [102]. SNSM has one JPCA II and one JPRA II that should be considered as recovery areas since it is known the area still harbors a jaguar population [8] and have active conservation actions and interesting market-based incentive programs (i.e., Jaguar Friendly). However, given the high transformation of the region [103], connectivity with other core habitats is practically null, which therefore affects genetic connectivity and hence increases the risk of local and regional extinctions, as previously suggested [20, 104]. In the case of Paramillo JPCA I, it is a protected area considered critical for ensuring the connectivity between the Caribbean and Pacific regions [20], and also as part of the continuum between Central and South America. Large scale and intensive conservation actions are urgently needed, especially by governmental institutions [70], for preserving the species and reducing current habitat transformation rates in the region.

Our study represents the first systematic approach proposing a connectivity network between remaining core habitat patches at national level. Previous efforts proposed 13 corridors between JCUs [31], however, our spatially-explicit approach further identifies 960 additional corridors distributed mainly in the Andean and Caribbean regions, which means a much more significant challenge for the species protection in Colombia than previously thought. For instance, corridors within the Andean region are critical for connecting the Amazon and Orinoco with the Pacific and Caribbean regions, but these corridors are of special concern considering they are immersed in highly transformed landscapes with high human densities [31, 96]. Additionally, a large proportion of corridors are located along the Magdalena River Basin, one of the country's most important rivers [105], but also one of the most

complicated and conflictive areas in Colombia. However, mobilizing communities around the benefits provided by the river together with the creation of conservation strategies and actions can represent a unique opportunity of combining conservation and development goals framed in the most important watershed of the country [106].

Our prioritization portfolio also allowed us to identify priority areas for jaguar recovery and conservation. JPCAs are important for the long-term maintenance of the species due to their large size, low HFI and high potential for connectivity. For example, JPCAs I from the Amazon, Orinoco, Pacific, and Serrania of San Lucas regions represent areas large enough for maintaining viable populations of >50 individuals [29], which have been previously identified [8], making them among the most important areas for ensuring not only jaguar but other species survival for the long term [20, 23]. On the other hand, the JPRAs require more significant conservation attention since most correspond to historical habitat for jaguars, but are currently under severe anthropic pressure and are most likely isolated, such as SNSM [70]. Considering connectivity therefore seems warranted as a long-term strategy for the preservation of more isolated populations [28] and thus for the maintenance of healthy jaguar populations in the country. Even though small-sized JPRAs are not large enough to host jaguar populations, they are supremely important in this conservation framework, since they might function as stepping-stones [29] for long-range dispersal between larger suitable areas [107].

Given the importance of JPCAs and JPRAs for the conservation and recovery of the species, these areas should be explicitly considered in territorial planning; interesting examples exist for instance in Santa Marta territorial planning which incorporated jaguar habitats as part of the municipality's environmental determinants [70]. Furthermore, these areas should be included in the planning for new protected areas, restoration portfolios, compensation schemes and incentives such as ecosystem services, market-based schemes, among others [97, 108, 109]. For instance, the Jaguar Friendly initiative is a market-based incentive program that certifies production units that directly contribute to jaguar conservation in human-dominated landscapes, thus articulating conservation and development and promoting coexistence, and with very promising results in cacao, coffee and forestry plantations in Costa Rica, Colombia and Venezuela [110]. Finally, a comprehensive strategy that incorporates the maintenance and expansion of suitable habitats [102] and securing individuals' mobility and dispersal and the consequent genetic diversity [24, 111], while reducing hunting, conflict, traffic and other population pressures, will considerably contribute to reducing the overall extinction risk of the species [24]. Lastly, jaguar conservation is a national challenge that requires the commitment of society as a whole, and safeguarding the species could serve as the best indicator and signal that the country has hope for securing its most valuable asset: biodiversity.

## Supporting information

**S1 Table. Records used for species distribution modeling of *Panthera onca* in Colombia.**
(DOCX)

**S2 Table. Models constructed for species distribution modeling and corresponding parameters for *Panthera onca* in Colombia.**
(DOCX)

## Acknowledgments

This work was developed between ProCAT Colombia and Laboratorio de Ecología Funcional from Pontificia Universidad Javeriana. We appreciate the support, comments and review from

J. Nicolas Urbina, the support from LEF Lab mates and ProCAT Colombia staff and we thank J. Schipper for language editing and comments to the manuscript.

## Author Contributions

**Conceptualization:** María Camila Machado-Aguilera, Jairo Pérez-Torres, José F. González-Maya.

**Data curation:** María Camila Machado-Aguilera, Leonardo Lemus-Mejía, Jairo Pérez-Torres, Diego A. Zárrate-Charry, Andrés Arias-Alzate, José F. González-Maya.

**Formal analysis:** María Camila Machado-Aguilera, Leonardo Lemus-Mejía, Jairo Pérez-Torres, Diego A. Zárrate-Charry, Andrés Arias-Alzate, José F. González-Maya.

**Investigation:** María Camila Machado-Aguilera, Jairo Pérez-Torres, Diego A. Zárrate-Charry, Andrés Arias-Alzate, José F. González-Maya.

**Methodology:** María Camila Machado-Aguilera, Leonardo Lemus-Mejía, Jairo Pérez-Torres, Diego A. Zárrate-Charry, Andrés Arias-Alzate, José F. González-Maya.

**Project administration:** María Camila Machado-Aguilera.

**Supervision:** Jairo Pérez-Torres, José F. González-Maya.

**Writing – original draft:** María Camila Machado-Aguilera, Leonardo Lemus-Mejía, Jairo Pérez-Torres, Diego A. Zárrate-Charry, Andrés Arias-Alzate, José F. González-Maya.

**Writing – review & editing:** María Camila Machado-Aguilera, Leonardo Lemus-Mejía, Jairo Pérez-Torres, Diego A. Zárrate-Charry, Andrés Arias-Alzate, José F. González-Maya.

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
