## [Decision Letter · Decision Letter 0]

3 Jan 2024

PONE-D-23-32228Preserving the spots of the land: Jaguar (Panthera onca) distribution and priority conservation areas in ColombiaPLOS ONE

Dear Dr. González-Maya,

Thank you for submitting your manuscript to PLOS ONE. After careful consideration, we feel that it has merit but does not fully meet PLOS ONE’s publication criteria as it currently stands. Therefore, we invite you to submit a revised version of the manuscript that addresses the points raised during the review process.

Please submit your revised manuscript by Feb 17 2024 11:59PM. If you will need more time than this to complete your revisions, please reply to this message or contact the journal office at plosone@plos.org. Please include the following items when submitting your revised manuscript:A rebuttal letter that responds to each point raised by the academic editor and reviewer(s). You should upload this letter as a separate file labeled 'Response to Reviewers'.A marked-up copy of your manuscript that highlights changes made to the original version. You should upload this as a separate file labeled 'Revised Manuscript with Track Changes'.An unmarked version of your revised paper without tracked changes. You should upload this as a separate file labeled 'Manuscript'.

We look forward to receiving your revised manuscript.

Kind regards,

Dárius Pukenis Tubelis, Ph.D.

Academic Editor

PLOS ONE

“This work was partially funded and developed in ProCAT Colombia and Laboratorio de Ecología Funcional from Pontificia Universidad Javeriana. We appreciate the support, comments and review from J. Nicolas Urbina-Cardona and Ginna Gomez-Junco, the support from Lab mates and ProCAT staff and the language editing and comments of Jan Schipper.”

5. We note that Figures 1, 2, 3, 4, 5, and 6 in your submission contain [map/satellite] images which may be copyrighted. All PLOS content is published under the Creative Commons Attribution License (CC BY 4.0), which means that the manuscript, images, and Supporting Information files will be freely available online, and any third party is permitted to access, download, copy, distribute, and use these materials in any way, even commercially, with proper attribution. For these reasons, we cannot publish previously copyrighted maps or satellite images created using proprietary data, such as Google software (Google Maps, Street View, and Earth). For more information, see our copyright guidelines: http://journals.plos.org/plosone/s/licenses-and-copyright.

1. You may seek permission from the original copyright holder of Figures 1, 2, 3, 4, 5, and 6 to publish the content specifically under the CC BY 4.0 license. 

Additional Editor Comments:

Dear Dr José González-Maya,

Thank you for submitting your study for publication in PLOS ONE.

We have received two reviews that suggested "Minor Revision" for your submission.

Both reviewers considered your study as important, interesting and well done.

Reviewer 1 is mainly concerned with the lenght of the Introduction and Discussion, and with the number of figures.

I ask that you try to consider these and other questions presented by this reviewer, when you consider appropriate. His/her corrections are in an attached file.

Reviewer 2 also liked your study and present a few corrections, that you also can consider.

The text of your manuscript has some problems regarding the English and punctuation (mainly commas). Reviewers pointed out some mistakes, but it

would be good if you can ask the help of an Englich native speaker before submitting the corrected version of your manuscript.

Additionally, I ask that you try to follow my sugestions below. They involve formatting and other aspects of the submission.

Review by the Editor Dárius Tubelis:

Title.

"spots" is not a common term in the Landscape Ecology literature. Why not "remnants" ?

Abstract.

Introduction.

I do not consider that the Introduction is too long, as pointed out by R1. Its lenght and content are appropriate.

Line 75. For sure, start a new sentence for the objectives - "....Colombia. The objectives of this study....".

Material and Methods

Line 106 ahead. I would appreciate a paragraph, or some sentences, in which you provide information on the system of protected

areas found in Colombia. How many reserves occur ? What is the territorial extension that they protect in total , and in each ecoregion? Basic information on this aspect

of the country would be welcome here in this section, as you discuss about this later.

Line 155. "fragmented forest" could be any of these 3 forest ypes....

Lines 157 to 160. Something is wrong with the writing. It appears to have a contradiction here.

Lines 227-228. This is important, but did you mention this in the objectives and methods ?

Line 229. This caption appears to be too short. There is nothing more to add ? maybe: ...based on...

Also, this caption is there but where have you cited Fig 2 along the text ? I did not find. Please check this.

Line 245. The caption is there, but where did you cite Fig 3 along the text ? I could not find it.

Line 259. Again for Fig 4. An "Error!" appeared some times along the last paragraphs. Maybe, this refer to your figure citations ? If so, this error

was done during the pdf creation. Please check your text.

Discussion.

You will make the reading easier if you create sub-sections with sub-titles along the Discussion. This would help to avoid eventual

repetitions, and reduce its lenght. Six pages it a bit long. Please try to reduce it a bit, as also suggested by R1.

Conclusion.

Please review it to avoid repeating parts of the Discussion. Maybe you can make it shorter.

References

They appear to be ok, but please make a last check before sending the corrected version of your manuscript.

Line 681. The words should not be with intials in capitals. Please check this aspect for all refs.

Lines 789-790. Is the common name correct ? Also, scientific names in itlalics.

Line 799. brackets not in italics.

Figures.

Would it be possible to combine figures 1 and 2 in only one ? Figure 2 appears to bring little information, and the map is "clean" (it could have major cities, major rivers, etc..and/or this info of Fig 1).

Reviewers' comments:

Reviewer's Responses to Questions

**Comments to the Author**

1. Is the manuscript technically sound, and do the data support the conclusions?

Reviewer #1: Yes

Reviewer #2: Yes

2. Has the statistical analysis been performed appropriately and rigorously? 

Reviewer #1: Yes

Reviewer #2: Yes

3. Have the authors made all data underlying the findings in their manuscript fully available?

Reviewer #1: Yes

Reviewer #2: Yes

4. Is the manuscript presented in an intelligible fashion and written in standard English?

Reviewer #1: No

Reviewer #2: Yes

5. Review Comments to the Author

Reviewer #1: Minor changes and improvements in English are needed. Cut back in length especially Intro and Discussion and delete Conclusions. Some English review is suggested and tightening the writing. Some highlights suggested at the end in Discussion to enhance the potential and applicability of the paper. I am sure the authors will resolve all of these without much difficulty. Please see the attached document with additional suggestions and comments.

Reviewer #2: Dear authors and editors,

Thank you for approaching me to review this paper. The authors developed a jaguar distribution map and determined potentially important areas for its conservation (corridors and patches) to inform conservation planning. I think the methods and science behind it are good and the information is an important contribution to jaguar conservation. I think the paper can be published as it is. However, there are some issues with the writing and style which I recommend reviewing critically. I have suggested a few points below but encourage the authors to do a careful inspection of these issues throughout the manuscript.

Below are other general comments for the authors to reflect on.

1. In line 111, it would be good to clearly state what the “primary sources” were. The authors only mention an example, but a clear list of what these sources were would be useful. For example, we are not sure whether this includes records of predation events or hunting events. Were these records considered in the models? Panthera has collected good records of predation and hunting events in Colombia in case authors would find this useful. This is not a requirement for publication, but I take the opportunity to respectfully mention this for future consideration.

2. Would it be possible to elaborate a bit more on the two criteria used for filtering the data? It is clear that the authors used two filters (credibility and geographic precision), but it is not straightforward to understand when a record is deemed as low, medium or high. This is just a matter of perhaps adding a parenthesis or something like that just for the sake of clarity.

3. Line 170: seems like the word should be “die” not “died”

4. There is regularly an error message in the ms stating that “Error! Reference source not found” This must be a cross reference or something similar, so please check this error for publication.

5. Line 310, change “comma” for a period in “. For example,”.

6. Line 321: correct the typo in “meanly” for “mainly”.

7. Line 367: correct the typo in “reduces” for “reduce”.

8. Line 402: correct typo in “taken”.

9. Lines 417-422: organize ideas, maybe split into a couple of sentences for better understanding.

10. Line 420: correct typo in "meanly”.

11. Line 426, delete the comma.

12. Line 427-428: check flow, correct (e.g. period is not correctly used).

13. Line 432: avoid shortening words (spell out “That´s”).

14. Line 441: delete “in this way”, this is a new paragraph no need for this link.

15. Line 460: delete “a”.

16. Line 461: correct type “meanly”.

Congratulations!

6. PLOS authors have the option to publish the peer review history of their article (what does this mean?). If published, this will include your full peer review and any attached files.

Reviewer #1: No

Reviewer #2: **Yes: **Lain E. Pardo

---

## [Author Response · Author response to Decision Letter 0]

18 Feb 2024

Responses to reviewers

R/ We double checked and adjusted accordingly.

R/ We added our data to OSF as open access, and we added the text to the manuscript. Thanks for the suggestion.

1. Gonzalez-Maya JF, Machado-Aguilera MC. Jaguar (Panthera onca) distribution records for Colombia. OSF; 2024. doi:10.17605/OSF.IO/RKDS9

R/ Sorry for the confusion. There was not specific funding or grants associated to the project but it was hosted at the institutions mentioned in the text. We fixed on the submission system.

“This work was partially funded and developed in ProCAT Colombia and Laboratorio de Ecología Funcional from Pontificia Universidad Javeriana. We appreciate the support, comments and review from J. Nicolas Urbina-Cardona and Ginna Gomez-Junco, the support from Lab mates and ProCAT staff and the language editing and comments of Jan Schipper.”

R/ Thanks for the comment, we will keep the Statement as it is, and removed from the MS the funding information. Thanks for raising the comment.

5. We note that Figures 1, 2, 3, 4, 5, and 6 in your submission contain [map/satellite] images which may be copyrighted. All PLOS content is published under the Creative Commons Attribution License (CC BY 4.0), which means that the manuscript, images, and Supporting Information files will be freely available online, and any third party is permitted to access, download, copy, distribute, and use these materials in any way, even commercially, with proper attribution. For these reasons, we cannot publish previously copyrighted maps or satellite images created using proprietary data, such as Google software (Google Maps, Street View, and Earth). For more information, see our copyright guidelines: http://journals.plos.org/plosone/s/licenses-and-copyright.

1. You may seek permission from the original copyright holder of Figures 1, 2, 3, 4, 5, and 6 to publish the content specifically under the CC BY 4.0 license. 

R/ Thanks for the comment. We double checked and used available public spatial data from worldclim2 and Vector Map Level 0, both which are open access and compatible with CC BY 4.0 license as states in their webpage (License. The data are freely available for academic use and other non-commercial use…. Using the data to create maps for publishing of academic research articles is allowed. Thus you can use the maps you made with WorldClim data for figures in articles published by PLoS, Springer Nature, Elsevier, MDPI, etc. You are allowed (but not required) to publish these articles (and the maps they contain) under an open license such as CC-BY as is the case with PLoS journals and may be the case with other open access articles.). We added the corresponding citation to each figure.

R: / We double checked the reference list and replaced and added some additional references related to the reviewers’ comments.

Reviewer 1

This paper is a comprehensive analysis of the connectivity in jaguar habitat in Colombia, a welcome addition to the urgently needed body of knowledge about this species to secure its future. I find the paper reasonably well-written, although I would recommend tighten the text overall by about 20%, and I would also suggest ensuring proper English language use. Starting with the title, the phrase “Preserving the spots of the land” does not make much sense idiomatically. The land does not have spots as such. I would recommend something like “Preserving the spots and claws: Jaguar….” Or similar

R: / We significantly reduced the lenght and tighten the text. Thanks for suggestions with the title, and although it is clear that we are not pretending to use spots as a land feature literally, we would like to maintain the game of words, where spots reflect a relationship with jaguars but also connects with spots as a reference (“a particular place, area, or part” Merriam-Webster), therefore making an analogy for jaguar spots to reflect the texture of the land. However, we changed the title to only “Preserving the spots”.

Line 52: Says: “Its large habitat and prey requirements” but I believe the authors mean something like “its requirements of extensive areas of forest and availability of large prey” or something like that, although some studies have shown that jaguars can survive in relatively small areas and feed on relatively small prey (Miranda et al 2016 J. Nat. Hist., Harmsen et al. 2011, Mamm Biol). Please resolve

R: / We adjusted the writing, including the consideration of the rteviewer. 

Variously, authors overuse the word “the”

R/ Corrected.

Line 227-228: Please expand the discussion about the category and status of the 15% protected territory

R: / Details added.

Line 252: How do you justify that a distance of merely 0.31 km is sufficient to classify an independent corridor from others? What is your benchmark for this and why?

R/ We believe the text was misinterpreted. We are referring to the minimum length of a corridor identified. We changed the text to avoid the problem. 

Line 297: You say that jaguar occupied distribution has shrunk by 43%. In how much time? Since when? Any chances you can illustrate the speed of decline?

R/ This is a very interesting observation and suggestion, although it is not possible to know the exact time frame in which the reduction occurred, due to the lack of references layers for it< nevertheless, we added a couple sources to try to provide a perspective on the subject.

Lines 292-456: The discussion is extremely and unjustifiably long. Please reduce by 40% 

R/ We made the best effort to reduce it as much as possible without compromising its cohesion.

Lines 457-478: Conclusions read like a selected repeat of the discussion. If no justification, please delete. 

R/ Thanks for pointing it out, effectively, we deleted the conclusion to avoid repetition.

I see no need for Figure 1, which is generic and can be found elsewhere. I also believe that there may be too many maps. Can you condense into only 4 or 5?

R/ Eliminated.

The authors refer to habitat destruction as the main threat to jaguars. Other authors have identified direct killing and conflict with humans as the main threats, given that jaguars are known to be able to survive in suboptimal habitat if they are not hunted. I would like to see a bit of discussion regarding the synergy that these threats combine and what is the status of the other two. Preserving only the habitat may not be sufficient to secure jaguar populations, as per the empty forest paradigm and others. Please comment

R/We added comments related to the subject for addressing the comment.

Also, I believe it would be important for the authors to weigh in in the discrepancy between IUCN enlisting the species as NT, while Colombia (and many other countries) enlist the species as VU or even EN. Do the authors agree with IUCN? They only state the status of the species in IUCN and in Colombia with no other comment.

R/ Although we appreciate the comment, and indeed we believe it is an interesting (and already long) discussion, it is not the aim or scope of our manuscript, and it is only mentioned as context in the first part of the introduction. Furthermore, we do not provide any new information for the total range of the species so we could provide a new perspective or new insights into the discussion, thus it is not covered by our approach. Nevertheless, we added some details on the status of the populations at continental scale to expand the conservation status context.

Reviewer 2

Title.

"spots" is not a common term in the Landscape Ecology literature. Why not "remnants" ?

Abstract.

R/ Thanks for the comment, we would prefer to maintain the game of words, although remnants is the proper literal term. We agree “spots” is not a common term in Landscape Ecology, but is not the intended use for an appealing title. However, we changed the title to only “Preserving the spots”.

Introduction.

I do not consider that the Introduction is too long, as pointed out by R1. Its lenght and content are appropriate.

R/ Thanks for the comment!

Line 75. For sure, start a new sentence for the objectives - "....Colombia. The objectives of this study....".

R/ Changed.

Material and Methods

Line 106 ahead. I would appreciate a paragraph, or some sentences, in which you provide information on the system of protected areas found in Colombia. How many reserves occur ? What is the territorial extension that they protect in total , and in each ecoregion? Basic information on this aspect of the country would be welcome here in this section, as you discuss about this later.

R/ Details added, thanks for the comment.

Line 155. "fragmented forest" could be any of these 3 forest ypes....

R/ Thanks for the comment. We are conscious that fragmented forest could include the other forest types, nevertheless, in the national land cover deataset there is a specific class for fragmented forest, reason why we included it. 

Lines 157 to 160. Something is wrong with the writing. It appears to have a contradiction here.

R/ Thanks, we reworded for clarity.

Lines 227-228. This is important, but did you mention this in the objectives and methods ?

R/ Thanks for pointing out the omission, we added the estimation to the methods section.

Line 229. This caption appears to be too short. There is nothing more to add ? maybe: ...based on...

R/ We added some more details to the caption.

Also, this caption is there but where have you cited Fig 2 along the text ? I did not find. Please check this.

R: / Corrected.

Line 245. The caption is there, but where did you cite Fig 3 along the text ? I could not find it.

R: / Corrected.

Line 259. Again for Fig 4. An "Error!" appeared some times along the last paragraphs. Maybe, this refer to your figure citations ? If so, this error was done during the pdf creation. Please check your text.

R: / Corrected.

Discussion.

You will make the reading easier if you create sub-sections with sub-titles along the Discussion. This would help to avoid eventual repetitions, and reduce its lenght. Six pages it a bit long. Please try to reduce it a bit, as also suggested by R1.

R: / Thanks for the comment; we reduced the length of the discussion considerably, therefore we think dividing in subsections is no longer needed and would fragment the discussion unnecessarily.

Conclusion.

Please review it to avoid repeating parts of the Discussion. Maybe you can make it shorter.

R: / Completely agree, we deleted the whole section.

References

They appear to be ok, but please make a last check before sending the corrected version of your manuscript.

R: / We double checked all the references.

Line 681. The words should not be with intials in capitals. Please check this aspect for all refs.

R: / Corrected.

Lines 789-790. Is the common name correct ? Also, scientific names in itlalics.

R: / Corrected.

Line 799. brackets not in italics.

R: / Corrected

Figures.

Would it be possible to combine figures 1 and 2 in only one ? Figure 2 appears to bring little information, and the map is "clean" (it could have major cities, major rivers, etc..and/or this info of Fig 1).

R/ Thanks for suggestion, we followed it.

General Comments

In line 111, it would be good to clearly state what the “primary sources” were. The authors only mention an example, but a clear list of what these sources were would be useful. For example, we are not sure whether this includes records of predation events or hunting events. Were these records considered in the models? Panthera has collected good records of predation and hunting events in Colombia in case authors would find this useful. This is not a requirement for publication, but I take the opportunity to respectfully mention this for future consideration.

R: / Thanks for the comment and we appreciate the offering. Undoubtedly, this information would be super useful in future approaches; nevertheless, for our modeling approach we used validated records collected by the authors throughout the years. We changed the text to make it clearer: "3) our own data derived from validated records of direct observations, camera-trap data, and predation events with confirmation of the predator, among other."

Would it be possible to elaborate a bit more on the two criteria used for filtering the data? It is clear that the authors used two filters (credibility and geographic precision), but it is not straightforward to understand when a record is deemed as low, medium or high. This is just a matter of perhaps adding a parenthesis or something like that just for the sake of clarity.

R/ Details added.

Line 170: seems like the word should be “die” not “died”

R/ Corrected.

There is regularly an error message in the ms stating that “Error! Reference source not found” This must be a cross reference or something similar, so please check this error for publication.

R/ Corrected.

Line 310, change “comma” for a period in “. For example,”.

R/ Corrected.

Line 321: correct the typo in “meanly” for “mainly”.

R/ Corrected.

Line 367: correct the typo in “reduces” for “reduce”.

R/ Corrected.

Line 402: correct typo in “taken”.

R/ Corrected.

Lines 417-422: organize ideas, maybe split into a couple of sentences for better understanding.

R/ Corrected.

Line 420: correct typo in "meanly”.

R/ Corrected.

Line 426, delete the comma.

R/ Corrected.

Line 427-428: check flow, correct (e.g. period is not correctly used).

R/ Corrected.

Line 432: avoid shortening words (spell out “That´s”).

R/ Corrected.

Line 441: delete “in this way”, this is a new paragraph no need for this link.

R/ Corrected.

15. Line 460: delete “a”.

R/ Corrected.

16. Line 461: correct type “meanly”.

R/ Corrected.

---

## [Editor Report · Decision Letter 1]

27 Feb 2024

Preserving the spots: Jaguar (Panthera onca) distribution and priority conservation areas in Colombia

PONE-D-23-32228R1

Dear Dr. González-Maya,

We’re pleased to inform you that your manuscript has been judged scientifically suitable for publication and will be formally accepted for publication once it meets all outstanding technical requirements.

Kind regards,

Dárius Pukenis Tubelis, Ph.D.

Academic Editor

PLOS ONE

Additional Editor Comments:

Dear Dr González-Maya,

Thank for submtting the corrected version of your manuscript on the Jaguar in Colombia (PONE-D-23-32228R1).

I have appreciated your responses to revieweres and the changes done.

With this, your manuscript was improved, and I now consider that it can be accepted for publication in PLOS ONE.

In my last reading, I found a range of minor mistakes that have to be fixed.

Please do so during the proofs correction, or another final stage of the evaluation process.

These mistakes/problems are:

Please check again if your affiliations are correct and according to the Guidelines.

Line 40. JPCA and JPRA are not appropriate as keywords, as they could mean different things in other countries or studies. Please replace them by

words such as the family name and "forest". Also, Jaguar is repetitive; consider "corridor".

Introduction

Here and along the text, there should be a space between the numbers of references: example [2, 3] instead of [2,3]. Please fix this along the ms.

Line 53. This reference [19] is wrongly placed. You should cite the number [8] here, as the previous reference was [7].

You have to change the position of ref 19 in the References Section, and change all posterior numbers along the text and in the References section. Be carefull !!

Accordingly, the references in line 55 would need to be [9, 10]. And so on...

Methods

Line 136. Is "(M)" wright here ? It sounds strange. Please check this.

Line 151. it is better to replace "&" by "and". The same for line 152.

Line 180. Change to "we used".

Line 196. Change to [91, 93]. Join them.

Line 203. ranges??

Lines 208 to 214. You were using values (e.g. "3") and now you use words (e.g. four). It is correct ? Please change for uniformity.

Results

Table 2. Consider writing "Max" in full (Bio5).

Line 245. Change to "had sizes".

Lines 270, 271. Some problem with the figure citation here.

Discussion

Line 302. Change to "have disappeared".

References

You are using the DOIs in a wrong way. The correct is "https://doi.org/numbers and letters". Please check the Guidelines and recent issues.

Some DOIs are missing....

Figure 5. Some numbers are overlapped in the scale.

That is all.

If you fix this, you article likely will be accepted in definitive.

You have produced an interesting and important paper.

Thank you for considering PLOS ONE as home of your research.

Dárius Tubelis

---

## [Editor Report · Acceptance letter]

13 Mar 2024

PONE-D-23-32228R1 

PLOS ONE

Dear Dr. González-Maya, 

I'm pleased to inform you that your manuscript has been deemed suitable for publication in PLOS ONE. Congratulations! Your manuscript is now being handed over to our production team.

Kind regards, 

on behalf of

Dr. Dárius Pukenis Tubelis 

Academic Editor

PLOS ONE